# Subconcussive head impact exposure between drill intensities in U.S. high school football

Kyle Kercher[1], Jesse A. Steinfeldt[2], Jonathan T. Macy[1], Keisuke Ejima[3], Keisuke Kawata[4,5]*

1 Department of Applied Health Science, School of Public Health-Bloomington, Indiana University, Bloomington, Indiana, United States of America, 2 Department of Counseling and Educational Psychology, School of Education, Indiana University, Bloomington, Indiana, United States of America, 3 Department of Epidemiology and Biostatistics, School of Public Health-Bloomington, Indiana University, Bloomington, Indiana, United States of America, 4 Department of Kinesiology, School of Public Health-Bloomington, Indiana University, Bloomington, Indiana, United States of America, 5 Program in Neuroscience, College of Arts and Sciences, Indiana University, Bloomington, Indiana, United States of America

* kkawata@indiana.edu

**Data Availability Statement:** All relevant data are within the manuscript and its Supporting Information files.

**Funding:** Indiana University Office of Vice President for Research (to K. Kawata)

## Abstract

USA Football established five levels-of-contact to guide the intensity of high school football practices. The objective of this study was to examine head impact frequency and magnitude by levels-of-contact to determine which drills had the greatest head impact exposure. Our primary hypothesis was that there would be an incremental increase in season-long head impact exposure between levels-of-contact: *air*<*bags*<*control*<*thud*<*live*. This observational study included 24 high-school football players during all 46 practices, 1 scrimmage, 9 junior varsity and 10 varsity games in the 2019 season. Players wore a sensor-installed mouthguard that monitored head impact frequency, peak linear acceleration (PLA), and rotational acceleration (PRA). Practice/game drills were filmed and categorized into five levels-of-contact (*air*, *bags*, *control*, *thud*, *live*), and head impact data were assigned into one of five levels-of-contact. Player position was categorized into lineman, hybrid, and skill. A total of 6016 head impacts were recorded during 5 levels-of-contact throughout the season. In the overall sample, total number of impacts, sum of PLA, and PRA per player increased in a near incremental manner (*air*<*bags*<*control* = *thud*<*live*), where *live* drills had significantly higher cumulative frequency (113.7±17.8 hits/player) and magnitude [2,657.6±432.0 $g$ (PLA), and 233.9 ± 40.1 krad/s$^2$ (PRA)] than any other levels-of-contact, whereas air drills showed the lowest cumulative frequency (7.7±1.9 hits/player) and magnitude [176.9±42.5 $g$ (PLA), PRA 16.7±4.2 krad/s$^2$ (PRA)]. There was no significant position group difference in cumulative head impact frequency and magnitude in a season. Although there was no difference in average head impact magnitude across five levels-of-contact and by position group PLA (18.2–23.2$g$) and PRA (1.6–2.3krad/s$^2$) per impact], high magnitude (60-100$g$ and >100$g$) head impacts were more frequently observed during *live* and *thud* drills. Level-of-contact influences cumulative head impact frequency and magnitude in high-school football, with players incurring frequent, high magnitude head impacts during *live*, *thud*, and *control*. It is

https://research.iu.edu/about/leadership/index.
html, the Spinal Cord & Brain Injury Research Fund
from the Indiana State Department of Health (to K.
Kawata: ISCBIRF 0019939) https://www.in.gov/
isdh/, and Indiana University Women's
Philanthropy Council (to K. Kawata) https://
iufoundation.iu.edu/leadership-giving/womens-
philanthropy/index.html. Sponsors had no role in
the design or execution of the study; collection,
management, analysis, or interpretation of the
data; preparation, review, or approval of the
manuscript; or decision to submit the manuscript
for publication.

**Competing interests:** The authors have declared
that no competing interests exist.

important to consider level-of-contact to refine clinical exposure guidelines to minimize head impact burden in high-school football.

## Introduction

The long-term consequence of sport-related head injury is a complex public health issue with no concrete solution [1, 2]. Despite inherent risk of head injury in contact sports (e.g., American football, hockey, soccer), participating in these team sports, especially during developmental age, provides well-documented benefits, including higher levels of physical activity, improved mental health, and lower likelihood of smoking cigarettes and using illegal substances [3]. In 2017, in an attempt to promote a safer football environment, USA Football (the national governing body over amateur football) developed a modified version of tackle football to introduce kids to the sport by reducing the field size and number of players, as well as rule changes to increase activity, game play, and learning [4]. These types of safety modifications are further substantiated in high school, college, and professional football. For example, in 2018 kickoff rules were adjusted to reduce injuries due to high concussion incidence during kickoff plays [5], players must wear helmets that meet certain laboratory safety standards, and a hit to the head or neck area and blindside blocking are prohibited [6, 7]. Consistent with these adjustments, concussion and catastrophic injury rates have been reduced [8, 9]; however, subconcussive head impact exposure has proven more complex.

Subconcussive head impact is defined as a hit to the head that does not induce overt concussion symptoms [10]. These head impacts are most common in American football, where athletes can experience several hundred impacts with some exceeding 1,000 head impacts in a single season [11]. Evidence has emerged to indicate that both high school and college football players with frequent experience of subconcussive head impacts exhibit neuronal microstructural damage [12, 13], abnormal brain activation [14, 15], ocular-motor impairment [16], and elevation in brain-injury blood biomarkers [17]. One line of research suggests that long-term exposure to these hits is a key factor in developing neurodegenerative disorders later in life [18, 19]. Although USA Football (for high school) and the NCAA (for college) have eliminated practicing two times in the same day (two-a-days) to minimize head impact frequency, one study found that total head impact frequency during a summer camp increased by 26% [20]. Similarly, a policy change to reduce the number of preseason practices from 29 to 25 failed to reduce head impact frequency in college football players, with one team's cumulative head impacts increasing up to 35% [21]. These mixed results highlight the need to dissect football practices to increase our understanding of what type of contact drills and intensities cause the greatest frequency and magnitude of subconcussive head impacts.

USA Football has identified five levels-of-contact (*air*, *bag*, *control*, *thud*, *live*) that define the intensities and structure of football practices nationwide. The levels-of-contact were designed to guide effective practice schedules through a step-by-step approach to teach fundamental football skills [22]. The National Federation of State High School Associations implemented the levels-of-contact in their high school football practice guidelines beginning in 2014. However, it remains unknown whether, and to what extent, different levels-of-contact influence head impact frequency and magnitude in high school football players across an entire season and between position groups.

Therefore, we conducted a longitudinal observational study to examine cumulative head impact frequency and magnitude across different drill intensities in high school football

players over the course of a single season. Our primary hypothesis was that there would be an incremental increase in season-long head impact frequency and magnitude between levels-of-contact, with *live* recording the greatest head impact exposure and air recording the lowest: *air < bags < control < thud < live*. Since the proximity to opponents and nature of contact during nearly every play for linemen [23], we also tested our secondary hypothesis that there would be a group difference in head impact frequency, in which linemen would have greater head impact frequency in most levels-of-contact, compared to the hybrid and skill positions. Our exploratory aim was to identify the average head impact magnitudes by levels-of-contact and position group, as well as to identify frequency of head impacts that were within 25–60 *g*, 60–100 *g*, or >100 *g* in each level-of-contact.

## Methods

### Participants

This single-site, observational study included 24 male high school football players at Bloomington High School-North. The study was conducted during the 2019 football season including practices and games during the pre-season, in-season, and playoffs. None of the 24 players was diagnosed with a concussion during the study period as confirmed by team athletic trainer and physician. Inclusion criterion was being an active football team member which was defined as any player, freshmen through seniors, planning to participate in the 2019 season. Exclusion criteria included a history of head neck injury (including concussion) in the previous year or any neurological disorders, although no participant met any exclusionary criteria. The Indiana University Institutional Review Board and the Monroe County Community School Corporation Research Review Board approved the study, and all participants and their legal guardians gave written informed consent.

### Study procedures

At the preseason data collection, self-reported demographic information (age, race/ethnicity, height, weight, number of previously diagnosed concussions, and years of experience in various contact sports, including tackle football) were obtained. Participants were custom-fitted with the Vector mouthguard (Athlete Intelligence, Inc.) that measured the number of hits and magnitude of head linear and rotational acceleration. Participants wore the mouthguard for all practices (n = 46), scrimmage (n = 1), and all games (n = 9 junior varsity, n = 10 varsity) from pre-season training camp (August 13, 2019) to the end of the season (November 1, 2019). The mean (SD) practice duration was 105 (20.5) minutes in duration. Video data were collected using Hudl (Agile Sports Technologies, Inc.) during the same timeframe as subconcussive head impact data collection. Participants' playing positions were verified by team coaches and categorized into three groups as follows: 11 linemen athletes (defensive lineman, offensive lineman), 7 hybrid athletes (tight end, linebacker, running back), and 6 skill athletes (wide receiver, defensive back), which is in line with prior literature [24, 25]. No quarterbacks participated in this study. In accordance with USA Football guidelines [26], head impacts were categorized by levels-of-contact: *air*, *bags*, *control*, *thud*, and *live*. See Levels-of-Contact and Film Review section for more details and supplemental file A for example video for each level-of-contact.

### Head impact measurement

This study used an instrumented Vector mouthguard for measuring frequency of head impacts as well as linear and rotational head accelerations during impacts, as previously described [27].

The mouthguard employs a triaxial accelerometer (ADXL377, Analog Devices) with 200 *g* maximum per axis to sense linear acceleration. For rotational acceleration, a triaxial gyroscope (L3GD20H, ST Microelectrics) was employed. An impact is detected when a linear acceleration magnitude exceeds 10.0 *g* for three consecutive samples (sampling every 0.2 milliseconds). All impact with a standard hit duration of 96 milliseconds were transmitted wirelessly through the antenna transmitter to the sideline antenna and computer, then stored on a secure internet database. The Vector mouthguard is installed with an in-mouth sensor to ensure that data acquisition occurs only when the mouthguard is securely fitted in one's mouth. Linear acceleration data were transformed within the Athlete Intelligence software to the head's center of gravity based on the 50th percentile male. From raw impact data extracted from the server, the number of hits, peak linear acceleration (PLA) of each hit, and peak rotational acceleration (PRA) of each hit were used for analyses. Kinematic accuracy of the prototype of Vector mouthguard [16] showed an excellent correlation with the matched data from an anthropomorphic testing device (crash test dummy) [28, 29]. When the mouthguard, headgear-mounted, and skin-patch sensors were compared to high speed video during soccer headings, the mouthguard showed superior skull coupling (displacement < 1 mm) compared to headgear (< 13mm) and skin patch (< 4mm) for the ear canal reference point [30]. A researcher was present during all practices to track when the practice shifted between levels-of-contact, and head impact data were categorized into each level-of-contact by corresponding time-stamps of head impact to timeframes of each level-of-contact.

## Levels-of-contact and film review

The five levels-of-contact are *air*, *bags*, *control*, *thud*, and *live* with *air* being estimated to have the lowest intensity and *live* being the highest [26]. *Air* is defined as drills being run unopposed and without contact. *Bags* is defined as drills being run against a bag or soft-contact surface. *Control* is defined as drills being run at an assigned speed until the moment of contact. It does not involve tackling, rather contact is above the waist and players stay on their feet. *Thud* is defined as drills being run at a competitive, fast speed through the moment of contact. It does not involve full tackling, rather contact is above the waist and players stay on their feet and a quick whistle ends the drills. *Live* is defined as drills being run in game-like conditions that include live-drill during practice as well as real games. *Live* should be the only time players are allowed to fully tackle another player to the ground. All head impacts in *air*, *bags*, *control*, and *thud* were during practices, whereas *live* occurred in both practices and games.

## Statistical analysis

Our primary aim was to examine whether cumulative head impact frequency, PLA, and PRA from a single season differ between 5 levels-of-contact (*air*, *bags*, *control*, *thud*, and *live*). Our secondary aim further examined the difference in these head impact measures between 3 position groups (lineman, hybrid, and skill). Three-way repeated measures ANOVA models were used to compare outcome variables (season-long cumulative head impact frequency, PLA, and PRA normalized per player) on 5 levels-of-contact and 3 groups. The assumption of sphericity was assessed with Mauchly's test and resulted in violation of sphericity (p<0.01), thus the Greenhouse-Geisser correction was used to report within-subject outputs, followed by the effect size (Partial Eta Squared). When there was a significant effect for levels-of-contact and/or group, then Bonferroni post-hoc tests were used to determine where the difference in head impact outcome occurred. For the exploratory aim, we similarly assessed whether average head impact magnitude differed by levels-of-contact and group using repeated measures ANOVA. Lastly, a total number of head impacts within ranges of 25–60 *g*, 60–100 *g*, or >100 *g*

in each level-of-contact was descriptively assessed in the overall sample. These exploratory thresholds are modeled based on the published papers that suggested $< 25$ $g$ as minimal magnitude [11, 31], 60 $g$ being previously thought to be a cut-off threshold to induce concussion [32], and 100 $g$ (precisely, 102.5 + 33.8 $g$) being an average magnitude leading to concussion diagnosis [24, 33]. All the data were analyzed using SPSS Statistics Version 25, and the level of statistical significance was set to p<0.05. Data are presented per player.

## Results

### Demographics and overall head impact exposure

A total of 6016 head impacts were recorded during 5 levels-of-contact in 24 high school football players throughout the season, resulting in a median of 203 hits, 4310.5 $g$, and 415.5 krad/$s^2$. Consistent with previous reports [13, 34], the distribution of head impact count was strongly right skewed with a median PLA of 19.7 $g$ (interquartile range: 15.3–27.8 $g$) and PRA of 1.8 krad/$s^2$ (interquartile range: 1.2–2.6 krad/$s^2$) per impact (Fig 1). These data are not reflective of head impacts that occurred outside the 5 levels-of-contact, such as walk-through and pre-practice/game conditioning. For comparison purposes, a driver can experience 30 $g$ to 40 $g$ of head and chest acceleration when a car collides into a fixed wall at 30 mph [35, 36]. Demographics and head impact data in the overall sample are detailed in Table 1.

### Level-of-contact-dependent cumulative head impact exposure

Levels-of-contact displayed an influence on cumulative head impact frequency and sum of PLA and PRA sustained during a season, as illustrated by a statistically significant main effect and medium effect size on levels-of-contact in the overall sample [Frequency, $F(1.95, 40.88) = 22.44$, p<0.001, $\eta_p^2 = 0.517$; PLA, $F(1.77, 37.08) = 21.70$, p<0.001, $\eta_p^2 = 0.508$; PRA, $F(1.62, 34.15) = 20.56$, p<0.001, $\eta_p^2 = 0.494$]. Bonferroni post-hoc tests revealed incremental increases in head impact frequency and sum of PLA and PRA as levels-of-contact intensify, except for between control and thud (*air* < *bags* < *control* = *thud* < *live*). For example, a football player experienced an average of 113.7±17.8 hits, 2,657.6±432.0 $g$ (PLA), and 233.9 ± 40.1 krad/$s^2$ (PRA) during *live* drills throughout a season, whereas 7.7±1.9 hits, 176.9±42.5 $g$ (PLA), PRA 16.7±4.2 krad/$s^2$ (PRA) during *air* drills throughout a season. See Fig 2A–2C for the visual trend of the outcomes and S1 Table for Bonferroni post-hoc results.

While all 3 position groups exhibited similar incremental patterns of head impact frequency and magnitude (*air* < *bags* < *control* < *thud* < *live*), there was no significant group difference in cumulative frequency as well as sums of PLA and PRA (Fig 3A–3C). Head impact kinematics for each level-of-contact are detailed in S2 Table.

### Head impact magnitudes by levels-of-contact and position group

Median head impact magnitudes (PLA and PRA) were similar across all levels-of-contact and all groups, ranging between 18.2 and 23.2 g for PLA (Fig 4A) and 1.6 and 2.3 krad/$s^2$ for PRA (Fig 4B) per head impact. See S3 Table for median PLA and PRA in each level-of-contact. Consistent with published papers [13, 23, 34], it was evident that a large number of head impacts across all levels-of-contact fell within 10 to 30 $g$, which might have diluted the minority of high magnitude head impacts. Our exploratory descriptive analysis identified that levels-of-contact also influence the number of high magnitude head impacts, whereby 60–100 $g$ and >100 $g$ were most prevalent in *live*, followed by *thud* and *control*, whereas very few hits were observed in *bags* and *air* (Fig 5A–5C).

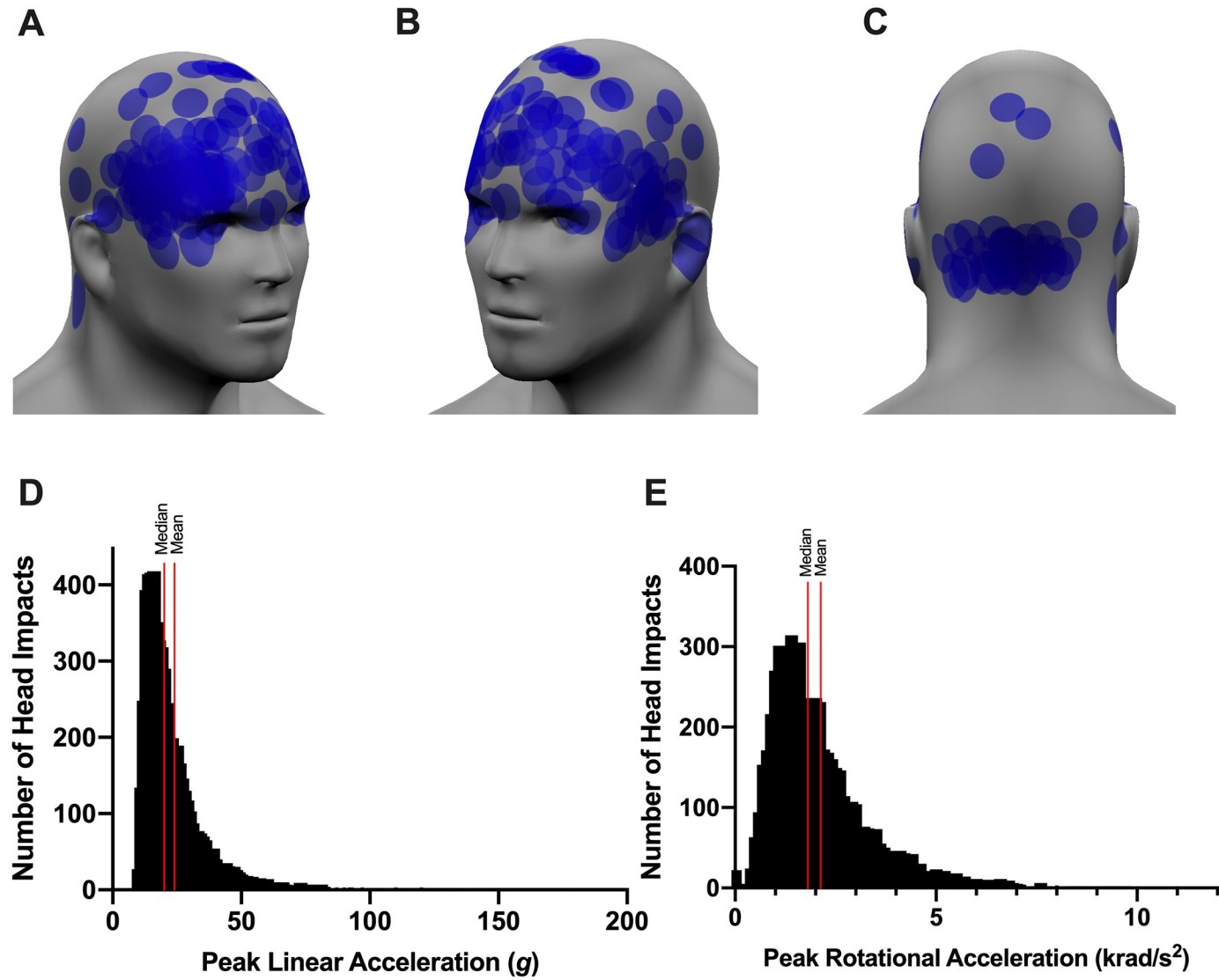

**Fig 1. Head impact distribution in 24 high school football players in a single season.** A representative data from a defensive lineman demonstrates the estimated locations of hits based on the data from the Vector mouthguard (A, front right; B, front left; C, back). Histogram of distribution of peak linear acceleration (D) and peak rotational acceleration (E) for all hits occurred during 5 levels-of-contact.

## Discussion

Subconcussion research is still at its infancy, but it is a rapidly growing area of concern in sport injury prevention. To contribute to this emerging field, this study examined whether cumulative subconcussive head impact frequency and magnitude in a single season differed across levels-of-contact and between player position groups. There were four key findings in this study. First, cumulative head impact frequency and magnitude increased as the level-of-contact increased, with the greatest head impact burden observed during *live*, followed by *thud* and *control* drills, and minimal head impacts during *bags* and *air* drills. Second, there were notable position group differences in head impact measures, where median values of head impacts in the linemen and hybrid players were greater in all levels-of-contact than those of the skill players. However, this was not supported by statistically significant group differences

**Table 1. Group demographics and head impact kinematics.**

| Variables | Overall | Linemen | Hybrid | Skill |
|---|---|---|---|---|
| N (%) | 24 (100) | 11 (46) | 7 (29) | 6 (25) |
| Age, y | 15.7 ± 1.1 | 16.1 ± 0.9 | 15.1 ± 1.2 | 15.7 ± 1.0 |
| BMI, kg/m$^2$ | 27.3 ± 6.3 | 31.6 ± 7.0 | 24.4 ± 2.5 | 23.0 ± 0.8 |
| No. of previous concussion | | | | |
| 0, $n$ (%) | 16 (66.7) | 6 (54.5) | 0 (0) | 3 (50.0) |
| 1, $n$ (%) | 6 (25.0) | 4 (36.4) | 0 (0) | 2 (33.3) |
| 2, $n$ (%) | 2 (8.3) | 1 (9.1) | 0 (0) | 1 (16.7) |
| Tackle football experience, y | 4.9 ± 2.7 | 5.8 ± 2.7 | 3.6 ± 2.5 | 4.8 ± 3.4 |
| Race, $n$ (%) | | | | |
| White | 21 (88) | 9 (82) | 7 (100) | 5 (83) |
| Black/African American | 0 (0) | 0 (0) | 0 (0) | 0 (0) |
| Asian | 0 (0) | 0 (0) | 0 (0) | 0 (0) |
| American Indian/Alaska | 1 (4) | 0 (0) | 0 (0) | 1 (17) |
| Multiracial | 2 (8) | 2 (18) | 0 (0) | 0 (0) |
| Ethnicity, $n$ (%) | | | | |
| Not Latino/Hispanic | 20 (83) | 9 (82) | 5 (71) | 6 (100) |
| Latino/Hispanic | 4 (17) | 2 (18) | 2 (29) | 0 (0) |
| Impact Kinematics for season, median (IQR) | | | | |
| Median cumulative impact count | 203 (118.0–350.0) | 213 (139.0–478.0) | 204 (153.5–314.5) | 131.5 (51.3–263.5) |
| Median cumulative peak linear acceleration, $g$ | 4310.5 (2686.3–8616.8) | 4289.5 (3178.6–11143.9) | 5262.7 (3445.1–7203.1) | 2837.7 (1117.2–6269.6) |
| Median cumulative peak rotational acceleration, krad/s$^2$ | 415.5 (179.2–828.0) | 358.7 (203.2–919.3) | 438.6 (293.0–655.9) | 296.5 (132.6–569.6) |

Data are reported as either mean (SD) or $n$ (%), except for head impact data using median (IQR). BMI, body mass index. IQR, interquartile range. OL, offensive lineman. DL, defensive lineman. TE, tight end. LB, linebacker. RB, running back. WR, wide receiver. DB, defensive back.

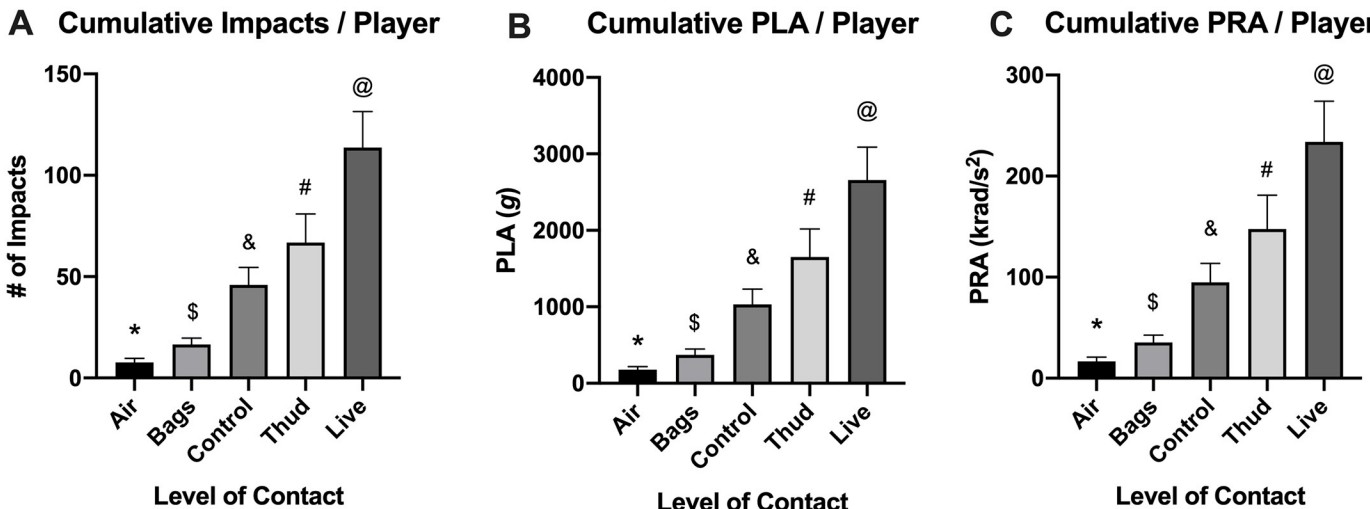

**Fig 2. Cumulative head impact kinematics between levels-of-contact throughout a season.** Cumulative (A) head impact count, (B) peak linear acceleration, and (C) peak rotational acceleration per player was influenced by the level-of-contact in an incremental manner, with *live* being the highest and *air* being the lowest. Data are presented as mean ± SD. Bonferroni post-hoc results are listed below: please refer to S1 Table for exact p-values of all possible comparisons. Please refer to S2 Table for median (IQR) values of head impact kinematics. @ Live is greater than thud (p = 0.045), control (p<0.001), bags (p<0.001) and air (p<0.001). # Thud is lesser than live (p = 0.045), no difference from control (p = 0.095), and greater than bags (p = 0.013) and air (p = 0.008). & Control is lesser than live (p<0.001), no difference from control (p = 0.095), and greater than bags (p = 0.008) and air (p = 0.003). $ Bags is lesser than live (p<0.001), thud (p<0.01), and control (p = 0.008), and greater than air (p = 0.041). * Air is lesser than live (p<0.001), thud (p = 0.007), control (p = 0.003), and bags (p = 0.041).

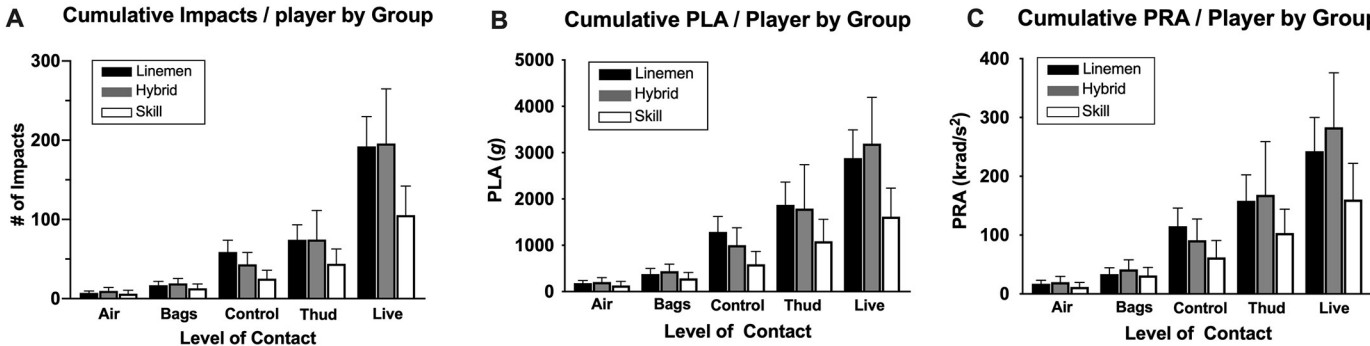

**Fig 3. Group-dependent cumulative head impact exposure between levels-of-contact.** All 3 groups shared similar incremental pattern in cumulative (A) head impact count, (B) peak linear acceleration, and (C) peak rotational acceleration per player, with *live* being the highest and *air* being the lowest. There was no group difference in the cumulative head impact frequency or magnitude. Data are presented as mean ± SD. Please refer to S2 Table for median (IQR) values of head impact kinematics.

likely due to the lack of sample size in each group. Third, the mean head impact magnitude was similar (18 to 23 *g*) across all levels-of-contact. Lastly, very high impact magnitudes (>100 *g*) were small in number overall but were more frequent in *thud* and *live* than other levels-of-contact. Taken together, our data, for the first time, empirically support the USA football's categorization of levels-of-contact while calling for a need to dissect football practice guidelines to make more specific recommendations for practice and games to minimize cumulative head impact burden on adolescents' brain health.

Owing to the sensor-installed helmets, mouthguards, headbands, and skin patches, our knowledge of head impact exposure in American football has drastically improved in the past 15 years. These technological advancements allowed researchers to evaluate head impact frequency and magnitude in various position groups, practice types (e.g., shell-only, full-gear),

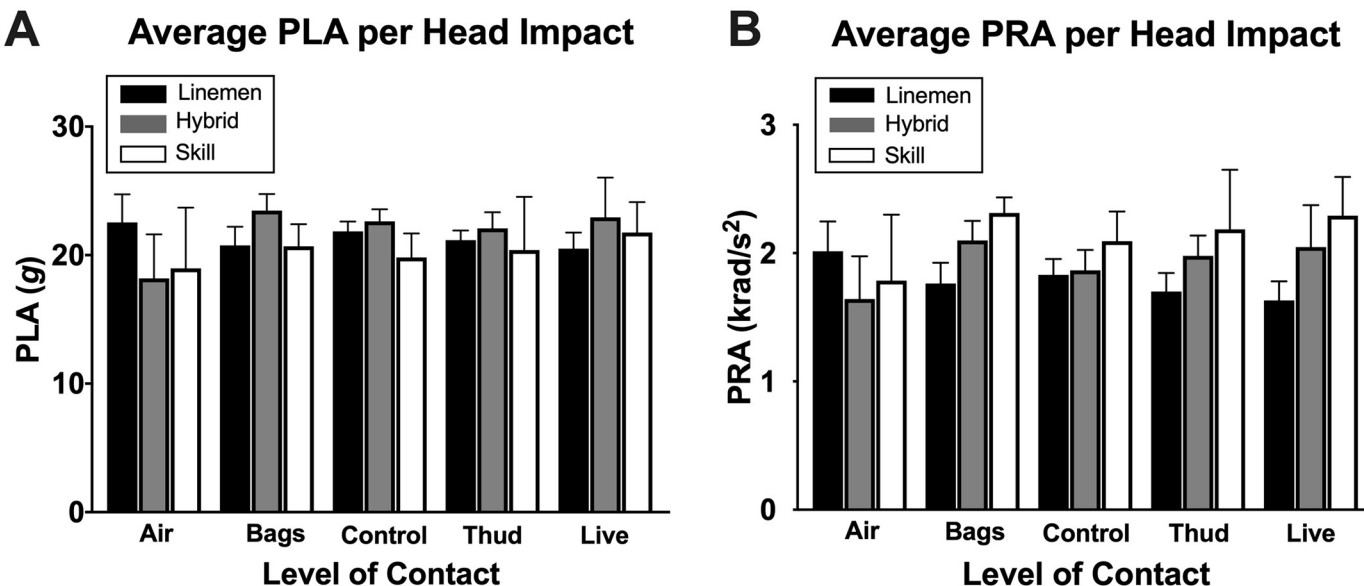

**Fig 4. Average head impact magnitude per impact between levels-of-contact and group.** There was no significant difference in average peak linear acceleration (A) and peak rotational acceleration (B) across 5 levels-of-contact and 3 groups. Data are presented as mean ± SD. See S3 Table for median (IQR) values of head impact kinematics.

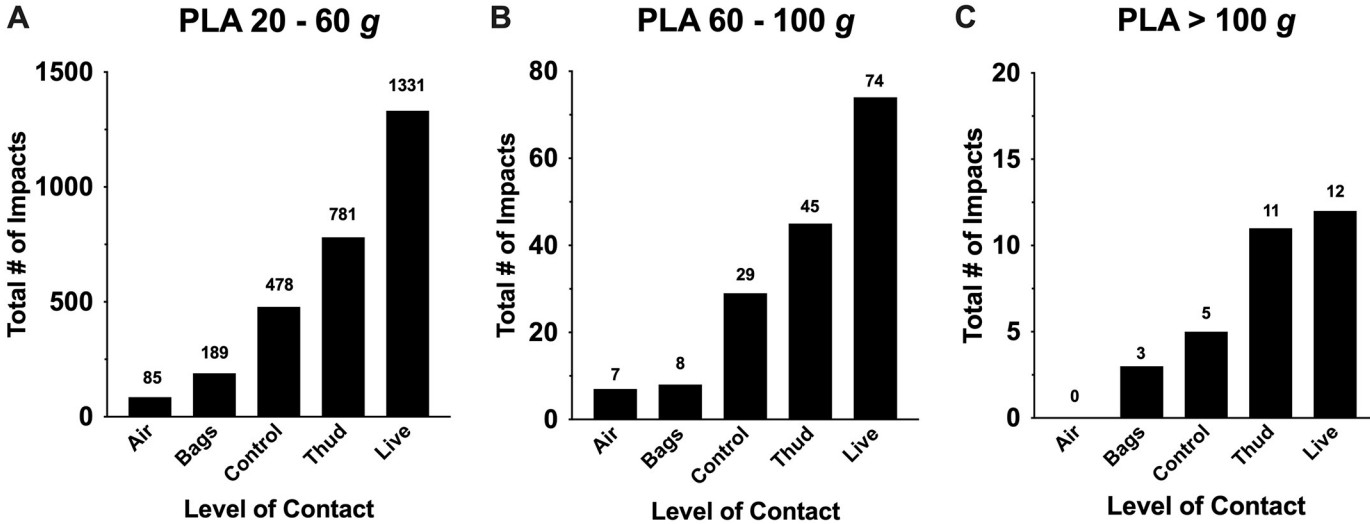

**Fig 5. Frequency of head impacts within various magnitude range.** A total number of head impacts from the overall sample throughout a season was categorized into peak linear acceleration ranging (A) 20–60 $g$, (B) 60–100 $g$, and (C) > 100 $g$. *Live* and *thud* consistently showed frequency head impacts in high impact magnitudes.

play types (e.g., running, passing, special teams), and time-based hit rates [37–43]. Previous research has suggested that overall head impact exposure elevates in relation to increased practice duration, contact intensity, and time spent in high risk drills [21, 39]. However, these variables differ greatly between players and position groups. For example, previous literature demonstrated that the differences between individual players accounted for 48% of the variance in head impact exposure during practice [44]. Additionally, recent research suggests different types of plays (e.g., running, passing, special teams) also have different average head impact magnitudes which will influence cumulative head impact exposure [41]. Despite this high degree of variance between players and play types, previous studies reported that the linemen and hybrid position groups consistently have higher head impact exposure compared to skill positions such as receivers, defensive backs, and quarterbacks [37, 38]. In our sample, we were able to observe a similar trend in the context of five different levels-of-contact, where the linemen and hybrid positions sustained a minimum of 27% more head impacts in *live* drills, with upwards of 3 to 4 fold higher head impact exposure in *air*, *bags*, and *control* than those of the skill position (see S2 Table). However, these group differences did not reach statistical significance likely due to the heterogeneous head impact exposure within each group (e.g., some skill players sustain many hits while several linemen experienced less hits), pointing to the issue of lack of sample size.

Another important finding from this study was that the average head impact magnitude (PLA and PRA) was similar (18 to 23 $g$) across 5 levels-of-contact. However, the frequency of strong magnitudes of head impacts (60–100 $g$ and >100 $g$) were greater during *live* and *thud* drills than other drills. These observations illustrate the fact that the majority of head impacts in high school football are considered mild, but the minority high magnitude impacts are evident in contact-prone drills. This evidence fills the critical gap in knowledge that the restriction of practice frequency and shortening of a season may not be effective unless considering the intensity of practice drills. Nevertheless, there have been attempts by various football governing bodies to restrict the amount of full contact in practices [20, 21, 26, 38, 45]. For example, USA Football's National Practice Guidelines for Youth Tackle Football suggest that

full-contact should not be done for more than 30 minutes per day and no more than 90 minutes per week during the regular season [26]. By limiting full contact practices from 3 days to 2 days per week, there was an average decline of 42% in head impact frequency (~250 hits per player) in high school football players [38]. In the same study, although the overall average head impact magnitude remained unchanged before and after reduction of contact practice, researchers found that a frequency of high magnitude head impacts were elevated in linemen and hybrid positions [38], suggesting that the team might have implemented higher intensity drills (i.e., *live*, *thud*, *control*) to compensate for the reduction of practice frequency. This compensatory trend was more conspicuous in the college setting. Even though the NCAA eliminated two-a-day practices and reduced practice frequency from 29 to 25 practices during summer camp, to compensate for the loss of practice times, football teams tended to incorporate more high-intensity, contact-prone practice drills, leading to increased head impact frequency [20, 21]. These data further substantiate the importance of regulating the duration of specific drill types, rather than restricting practice type and frequency.

Several studies suggest that football players who go on to sustain concussion tend to experience frequent subconcussive head impacts [24, 33]. Since none of the players in the current study were diagnosed with concussion, we were unable to suggest the potential preventive effect of levels-of-contact on concussion incidence. However, it is noteworthy that our exploratory analysis revealed no correlation (r = 0.07, p = 0.71) between previous number of concussion and subconcussive head impacts sustained during this season. This finding suggests that previous concussion history has almost no influence in how players perform, in the context of head impact exposure, during the season.

## Clinical implications

Pending confirmation by a larger-scale study, these results may have important implications for the clinical management and guidelines regulating subconcussive head impact exposure in high school football. If a restriction of contact-prone drills (i.e., live, thud) during practice is indeed effective in reducing head impact frequency and associated magnitude, establishing a policy or guideline to minimize head impact burden would be a logical next step. In addition to the consideration of level-of-contact, the most effective method for informing clinical guidelines may be a blend of the different strategies. For example, it is important to implement guidelines informed by head impact data derived not only from entire teams or position groups, but also from more targeted and specific practice structural variables such as drill type and hit per player per minute rates. Additionally, individual-level variables such as tackling technique, starting status, or count of repetitions players participate in would be imperative to determine who might be at risk for sustaining many head impacts in a short window. Fortunately, coaches across the nation have the ability to implement the USA Football levels-of-contact within their practice structures with relatively minimal effort. This strategy may allow coaches to directly influence the total subconcussive head impact exposure in their athletes.

## Limitations

There are several limitations to this study. Our examination of head impacts in high school football is limited in that it was conducted on a single high school football team in the Midwest composed of primarily white males. Because of lack of racial and ethnical diversity in our sample, we were unable to conduct any analysis to identify whether race/ethnicity played a role in subconcussive head impact exposure. This should be addressed in a future study along with whether race/ethnical background influence one's neurologic resiliency and susceptibility to subconcussive head impacts. Therefore, the results from the current study are not

generalizable to the broader U.S. population of high school football teams. A second limitation of the study is that the USA Football levels-of-contact guidelines are just that, guidelines. They are not legislation, and it is unknown what percentage of high schools currently utilize the USA Football level-of-contact system; thus, this further limits the generalizability. Implementation of the levels-of-contact is not a perfectly reliable variable in that situations occur in practice that may have overlapping levels-of-contact. This overlap made cumulative calculations of time spent in each level-of-contact difficult during certain drills and the time spent in each level is likely to influence cumulative head impact exposure. Coaches in this high school attempted to implement the levels-of-contact consistently, but to control up to 22 athletes participating in a single drill at the same time is not always feasible. An additional limitation is that the Vector mouthguard has not been validated for true positive versus false positive data. These limitations portend a follow-up research question as to whether regulating the number of repetitions in each level-of-contact is a more effective approach than teamwide contact restrictions for minimizing the frequency and magnitude of subconcussive head impacts in individual players (or position groups). Hence, for future research we suggest examining subject-specific head impact data and playing style (run-first vs. pass-first offense) [46] that incorporate multiple predictor variables such as time spent in each level-of-contact, the repetitions per player per drill, levels-of-contact, position group, and impacts per event (i.e., drill, practice/game, or week) for head impact outcomes.

## Conclusion

Levels-of-contact may influence cumulative head impact frequency and magnitude in high school football players, with players incurring frequent head impacts during *live*, *thud*, and *control*. Strong magnitudes of head impacts ($> 60\ g$) were frequently observed especially during *live* and *thud*. It is important to consider levels-of-contact to refine football practice guidelines/policies to minimize cumulative head impact burden in high school football players.

## Supporting information

**S1 Table. Bonferroni post-hoc results on 5 levels of contact.**
(DOCX)

**S2 Table. Cumulative head impact frequency and magnitude for the entire season.**
(DOCX)

**S3 Table. Average peak linear acceleration and peak rotation acceleration.**
(DOCX)

**S1 Video.**
(MP4)

## Author Contributions

**Conceptualization:** Kyle Kercher, Jesse A. Steinfeldt, Jonathan T. Macy, Keisuke Kawata.

**Data curation:** Kyle Kercher.

**Formal analysis:** Kyle Kercher, Keisuke Ejima, Keisuke Kawata.

**Funding acquisition:** Keisuke Kawata.

**Investigation:** Kyle Kercher, Jesse A. Steinfeldt, Jonathan T. Macy, Keisuke Ejima, Keisuke Kawata.

**Methodology:** Kyle Kercher, Jesse A. Steinfeldt, Jonathan T. Macy, Keisuke Kawata.

**Project administration:** Kyle Kercher, Jesse A. Steinfeldt, Keisuke Kawata.

**Resources:** Kyle Kercher, Jesse A. Steinfeldt, Keisuke Ejima, Keisuke Kawata.

**Supervision:** Jonathan T. Macy, Keisuke Kawata.

**Visualization:** Kyle Kercher, Keisuke Kawata.

**Writing – original draft:** Kyle Kercher, Keisuke Kawata.

**Writing – review & editing:** Jesse A. Steinfeldt, Jonathan T. Macy, Keisuke Ejima, Keisuke Kawata.

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
