## [Decision Letter · Decision Letter 0]

3 Jul 2020

PONE-D-20-13120

Subconcussive head impact exposure between drill intensities in U.S. high school football

PLOS ONE

Dear Dr. Kawata,

Thank you for submitting your manuscript to PLOS ONE. After careful consideration, we feel that it has merit but does not fully meet PLOS ONE’s publication criteria as it currently stands. Therefore, we invite you to submit a revised version of the manuscript that addresses the points raised during the review process.

Due to the importance of your submitted manuscript for the sports community there has been a rigorous peer-review by different specialists in neuro-science and / or concussion researchers.

I am very pleased to tell you that all peer-reviewers found your manuscript valuable and methodological sound.

Nevertheless there are some minor revisions needed before I can consider the manuscript suitable for publication.

I am looking forward to act as an academic editor for your revised manuscript soon.

We look forward to receiving your revised manuscript.

Kind regards,

Markus Geßlein

Academic Editor

PLOS ONE

Journal Requirements:

Reviewers' comments:

Reviewer's Responses to Questions

**Comments to the Author**

1. Is the manuscript technically sound, and do the data support the conclusions?

Reviewer #1: Yes

Reviewer #2: Yes

Reviewer #3: Yes

Reviewer #4: Yes

Reviewer #5: Yes

2. Has the statistical analysis been performed appropriately and rigorously? 

Reviewer #1: Yes

Reviewer #2: Yes

Reviewer #3: Yes

Reviewer #4: I Don't Know

Reviewer #5: Yes

3. Have the authors made all data underlying the findings in their manuscript fully available?

Reviewer #1: Yes

Reviewer #2: Yes

Reviewer #3: Yes

Reviewer #4: Yes

Reviewer #5: Yes

4. Is the manuscript presented in an intelligible fashion and written in standard English?

Reviewer #1: Yes

Reviewer #2: Yes

Reviewer #3: Yes

Reviewer #4: Yes

Reviewer #5: Yes

5. Review Comments to the Author

Reviewer #1: Dear Editor,

Dear Authors,

Thank you very much for the opportunity to review the manuscript "Subconcussive head impact exposure between drill intensities in U.S. high school football."

The manuscript by Kercher et al. suggests that level-of-contact in high school football practices influences cumulative head impact frequency and magnitude. With the use of a sensor-installed mouthguard, head impact frequency, peak linear acceleration and rotational acceleration were investigated and categorized into five levels-of-contact (air, bags, control, thud, live) as well as three player positions (lineman, hybrid, skill). The total number of impacts per player increased in a near incremental manner, where live drills had higher cumulative frequency than the other levels-of-contact. High magnitude (60-100 g and >100 g) head impacts could be more frequently observed during live and thud drills compared with air, bag and control. The manuscript is well written and the main issue that authors discuss would be important for prevention or minimization of head impact burden in high school football by refining the clinical exposure guidelines.

This is an interesting and potentially valuable study but the manuscript could be improved in some ways. A number of minor critical points are described below:

Abstract

[P2, L29] The authors should add the number of days of all practices and games in the 2019 season.

Introduction

[P3, L51] A brief description of the long-term consequences of a sport-related head injury, especially in the context of (high school) football, would be useful.

Methods

[P5, L109] How was an active football team member defined? The authors should describe the inclusion criteria in more detail.

[P6, L118] What role does the determination of race/ethnicity play in the study and what conclusions can be drawn from the results? If these variables are not informative, they should be excluded.

[P6, L122] The observation period from pre-season training camp to the end of the season should be stated with the exact number of days and average minute duration on which the practice was performed.

[P6, L128] Why were no quarterbacks included in the study? The authors should state reasons for the exclusion.

[P8, L178] Why was the total number of head impacts split into groups of 25-60 g, 60-100 g or >100 g? The authors should add a short explanation why this categorization was chosen.

Results

[Table 1] The variables “Playing position, n (%)” seem to have the same meaning as the first variable "n". Perhaps the authors could simply put the percentage in brackets after the variable "n" to make the table clearer.

The authors might also consider referring to the following articles:

Crisco JJ, Wilcox BJ, Beckwith JG, et al. Head impact exposure in collegiate football players. J Biomech. 2011;44(15):2673‐2678.

Martini D, Eckner J, Kutcher J, Broglio SP. Subconcussive head impact biomechanics: comparing differing offensive schemes. Med Sci Sports Exerc. 2013;45(4):755‐761.

Reviewer #2: Thank you for the opportunity to review the manuscript titled “Subconcussive head impact exposure between drill intensities in U.S. high school football”. This study examined head impact frequency and magnitude by different levels of contact and player position in a small cohort of high school football players. The study is well conceived and conducted. The manuscript is well structured and written. I only offer a few minor comments that will hopefully assist the authors with improving the presentation of their findings.

1. For the subsection about ‘Head impact measurement’ (p. 6-7), please add some pertinent information about the reliability and validity of the measurement tool (i.e. Vector mouthguard).

2. For Table 1, the row with sex can be omitted. The proportions by position group can be moved to the row with sample sizes. It is unnecessary to report age, BMI, experience, PLA, and PRA using two decimal places. It gives a false sense of precision of the measurements. I think one decimal place will suffice.

3. For the subsection on ‘Average head impact magnitudes by levels-of-contact and position group’ (p. 11), the word ‘average’ is ambiguous. I suggest making is clearer that all the values are medians.

4. For Figure 2, please make it clear that the plots show median number of impacts per player. Presumably the error bars indicate the IQR. In the footnotes, please ensure that all p-values greater than 0.001 are reported with and equal symbol (i.e. not inequality symbol).

5. For Figure 3, please make it clear that the plots show median number of impacts per player. Presumably the error bars indicate the IQR.

6. For Figure 3, please make it clear that the plots show median number PLA and PRA. Presumably the error bars indicate the IQR.

Reviewer #3: Summary:

The authors provide an observational study of subconcussive head impact frequency and magnitude in a high-school-football team during the 2019 season depending on level-of-contact and player position during training and game sessions. They followed-up on a high-school football team in the mid-west of the USA including 24 male teenagers (age 15,7 +/- 1,08 y), with player positions lineman, hybrid and skill, during the whole 2019 season, including all training sessions. Written informed consent of players and legal guardians was given. The players wore sensor-installed mouthguards monitoring impact frequency, peak linear and rotational acceleration to every training session and game. Via video-analyses of all training sessions and games, activity was categorized into five levels of contact with increasing intensity, as established by USA Football (air, bags, control, thud, live).

The research question deals with whether and to what extent, different levels-of-contact and different player position influence head impact frequency and magnitude. The primary hypothesis suggests an incremental increase in head impact frequency and magnitude over the season between levels of contact (air<bags<control<thud<live). and="" differences="" frequency="" group="" head="" hypothesis="" impact="" magnitude="" regarding="" secondary="" suggests="" the="">

The authors give a clear objective and clear research questions, and these are up to date and relevantly embedded into the current literature. They address an important subject, as repetitive head injuries lead to structural damage and neurologic deterioration in athletes. Especially in children and young adults, grave consequences can arise as their brains are still developing. Moreover, the impact of subconcussive head injuries on the development of structural and neurobehavioral pathologies is not clear. Therefore, the systematic measurement of head impact frequency and magnitude in a high-school team during one season, as conducted in this study, is of relevance.

The figures and tables included in the study display the results comprehensively. The supplementary material helps visualizing head impact frequency and magnitude and highlights the study design. All statistic findings are displayed in the supplementary material comprehensively.

The authors collected 6016 head impacts throughout the season. Total number of impacts, magnitude and acceleration increased with level-of-contact (air<bags<control=thud<live). and="" difference.="" frequently="" group="" head="" high="" impacts="" in="" live="" magnitude="" more="" no="" position="" significant="" there="" thud="" was="" were="">

The results are comprehensively displayed, visualized, and supported by the figures and tables in the manuscript and the supplementary material. They are set into context to the recent relevant literature. The methods are appropriate and reproducible, in the supplementary material even video examples for analysis are included.

The authors conclude with the suggestion to include the 5 level-of-contact recommendations to refine clinical exposure guidelines to minimize head impact burden in high-school football. The conclusion matches the research question and the results and is clinically relevant.

Overall impression:

The authors provide a relevant, interestingly designed observational study with 24 teenage athletes over a whole high-school football season. They combine the technical measurement of head impact magnitude and acceleration data via sensor-installed mouthguards with video-analyzed levels-of contact and different player position. The research question is relevant and up to date, the hypotheses are well-chosen and the results and conclusion match. The study contributes to the steadily evolving body of evidence that further and stricter regulations in recreational and professional sport are necessary to reduce the risk of repetitive head impacts and their potentially grave consequences. We recommend publishing this work and give only minor issues, as mentioned below.

Major Issues:

None.

Minor Issues:

1. We would recommend including the research question and hypotheses in the Abstract.

2. We would also recommend giving the dates the mentioned rule-changes and recommendations for reducing head trauma frequency in high-school / college football were established (levels-of-contact, tackle football, kickoffs rules) for a better overview on the subject and current stand (lines 56-65 and 74-79).

3. The authors give the number of previous concussions in Table 1 (line 193), however, they do not describe in the manuscript the relevance of their presence or absence; we would recommend to also meet this topic of risk potentiation in repetitive head impacts.

4. The authors also give in Table 1 (line 193) in detail the race and ethnicity of the players, although 88% of the players are white and only 4% American Indian and 8% multiracial. Moreover, they do not explain in the manuscript potential interracial differences in concussion risk. We would recommend to either add this information or to shorten the table regarding this aspect.</bags<control=thud<live).></bags<control<thud<live).>

Reviewer #4: Dear authors,

thank you very much for this interesting study. I have some comments below.

Title

ok

Abstract

ok

Introduction

well written, good introduction to the topic, clear research question

Material and Methods

line 123: this is a short season from mid August 2019 to end of October 2019 for preseason training to the end of season?

Results

line 197: the results are clearly describing the level of contact and head impact exposure. However, what is missing, is how much time the athletes were performing activities of the different contact levels. e.g. if most time was spent doing live contact, it is no wonder that most head impact was generated during this typ of impact. it would be optimal to also include exposure time...

Discussion

line 231: also dependent on exposure time, see comment above...

line 233: the skill group received much less cumulative impacts 130 vs over 200. Although no statistically significant finding, don't you think that receiving 30% less impacts does not count? please discuss! otherwise please add power analysis and sample size calculation etc... please see also line 254 where you say so, that skill positions have leas head impact

line 299ff: another limitation is that exposure time of the different levels of contact is not reported...

line 318: usually references are not cited in the conclusion, since the conclusions are your main findings and not the findings of other studies (since you published an original study)

References

ok

Tables

Table 1:please remove psychiatric conditions, since there are none. makes the table smaller and easier to read... also remove the lines "playing position, n (%)", since this information is redundant. the cumulative impact count is actually "average cumulative impact count"? also for the peak linear and rotational acceleration?

Figures

ok

Reviewer #5: For me as a neurologist it is very important that efforts are made to reduce the cumulative exposure of subconcussional cerebral microtrauma in adolescent high school football players. To verify the validity of the concept of different levels-of-contact which has been establishes by USA Football, the authors present an interesting study that compares the real exposure to mild cranial trauma at different levels-of-contact and different positions of football players. In this respect, the authors prove and validate the intuitive assumption that the different increasing levels-of-contact reflect a hierarchy of cumulative exposure to hits to the head. I have no objections to or suggestions for improvement of the manuscript.

6. PLOS authors have the option to publish the peer review history of their article (what does this mean?). If published, this will include your full peer review and any attached files.

Reviewer #1: **Yes: **Juergen Taxis

Reviewer #2: No

Reviewer #3: No

Reviewer #4: No

Reviewer #5: No

---

## [Author Response · Author response to Decision Letter 0]

12 Jul 2020

We thank the reviewers and editor for their appreciation of the potential clinical significance of the current manuscript and their detailed and constructive comments, which have strengthened and clarified the methodology and deliverable of this manuscript. As the editor and all reviewers acknowledged, this study will play a significant catalyst to improve policy/guideline to ensure players’ safety. 

Reviewer #1: 

Abstract

The authors should add the number of days of all practices and games in the 2019 season.

RESPONSE: Thank you for the suggestion. We added the number of practices (46), scrimmage (1), junior varsity (9), and varsity games (10) to the abstract, as well as in the study procedure of the method section. 

Introduction

[P3, L51] A brief description of the long-term consequences of a sport-related head injury, especially in the context of (high school) football, would be useful.

RESPONSE: Excellent point. We have had a long discussion about this point prior to the initial submission whether to mention the potential long-term consequences (e.g., CTE, dementia). Although many data are beginning to point to the adverse long-term outcome, we are also aware of several opposing lines of research. This manuscript does not assess neurologic consequences. Instead, our objective was to examine head impact exposure by levels-of-contact; therefore, we made a conscious decision not to dive into long-term neurologic affects. We instead included some of the acute/subacute (e.g., season-long head impact) neurologic outcomes in the 2nd paragraph of the Introduction. We would like to maintain this stance if that is acceptable. If you think we need to include the potential long-term consequence (e.g., CTE), please let us know. 

Methods

[P5, L109] How was an active football team member defined? The authors should describe the inclusion criteria in more detail.

RESPONSE: Clarification of this was added to the first paragraph of the methods section.

[P6, L118] What role does the determination of race/ethnicity play in the study and what conclusions can be drawn from the results? If these variables are not informative, they should be excluded.

RESPONSE: This is an important point. We listed the lack of generalizability to the general population as the first weakness in our limitations section due in part to the small sample size and lack of racial and diversity (i.e., primarily white males). Because of lack of racial and ethnical diversity, we were unable to conduct any analysis to identify whether race/ethnicity played a role in head impact exposure (i.e., White players tend to sustain more/less head impacts than black and other race). However, we felt as though it is important to state that we were working with a primarily white sample because future research may have different findings/conclusions when working with different race/ethnicity samples. Especially because this is first of its kind to delineate level-of-contact influence on head impact exposure, we think it is important to report race/ethnicity data. We expanded this aspect in the limitation section. 

[P6, L122] The observation period from pre-season training camp to the end of the season should be stated with the exact number of days and average minute duration on which the practice was performed.

RESPONSE: Thank you for pointing this out. We added the exact number of observed practices and games along with the mean (SD) minute duration to the line you suggested.

[P6, L128] Why were no quarterbacks included in the study? The authors should state reasons for the exclusion.

RESPONSE: Excellent point to clarify. There was only one quarterback in the entire high school program during the study enrollment period, and unfortunately, he declined to participate in the study due to his lack of interest and reluctancy to wear the Vector mouthguard. We have clarified this point in the manuscript per your request.

[P8, L178] Why was the total number of head impacts split into groups of 25-60 g, 60-100 g or >100 g? The authors should add a short explanation why this categorization was chosen.

RESPONSE: Thank you for this comment. We clarified these exploratory thresholds in the line suggested, by explaining that we modeled Barber-Foss et al. 2019 and Bailes et al. 2013 that suggested < 25 g as minimal or negligible magnitude; Guskiewicz et al. 2007 that introduced 60 g as a cut-off threshold to induce concussion, whose nuance was softened after many papers reporting that concussion can occur head impact below 60 g. Nonetheless, we thought this was a relevant threshold to descriptively present our data. Lastly, Beckwith et al. 2013 reported that average head impact magnitude that led to concussion diagnosis was 100 g (precisely, 102.5 + 33.8 g). These points are now summarized in the revised manuscript.

Results

[Table 1] The variables “Playing position, n (%)” seem to have the same meaning as the first variable "n". Perhaps the authors could simply put the percentage in brackets after the variable "n" to make the table clearer.

RESPONSE: We have implemented this suggestion.

The authors might also consider referring to the following articles:

Crisco JJ, Wilcox BJ, Beckwith JG, et al. Head impact exposure in collegiate football players. J Biomech. 2011;44(15):2673‐2678.

Martini D, Eckner J, Kutcher J, Broglio SP. Subconcussive head impact biomechanics: comparing differing offensive schemes. Med Sci Sports Exerc. 2013;45(4):755‐761.

RESPONSE: We added the Crisco article as a reference as it pertains to suggesting significant variance in head impact exposure between position groups. We also included Martini et al paper to suggest the potential interaction between playing style (run-first vs. pass-first offense) and level-of-contact drills. Thanks for the suggestion. 

Reviewer #2: 

1. For the subsection about ‘Head impact measurement’ (p. 6-7), please add some pertinent information about the reliability and validity of the measurement tool (i.e. Vector mouthguard).

RESPONSE: Thank you for pointing this out. We added a comment in the limitations section about the lack of literature validating the Vector mouthguard for true positive versus false positive head impacts, although our unpublished internal validation from our previous study1,2 yielded 85-90% agreement between Vector mouthguard and film analysis. Additionally, we added a statement in the Head impact measurement section on the reliability of movement in sensor-installed mouthguards compared to skin patch and skull cap head impact devices.

1. Zonner S, Ejima K, Bevilacqua ZW, Huibregtse ME, Charleston C, Fulgar CC, Kawata K. Association of increased serum S100B levels with high school football subconcussive head impacts. Front Neurol. 2019;10:327.

2. Zonner SW, Ejima K, Fulgar CC, Charleston C, Huibregtse ME, Bevilacqua ZW, Kawata K. Oculomotor response to cumulative subconcussive head impacts in high school football players: a pilot longitudinal study. JAMA Ophthalmol. 2019;137(3):265-270.

2. For Table 1, the row with sex can be omitted. The proportions by position group can be moved to the row with sample sizes. It is unnecessary to report age, BMI, experience, PLA, and PRA using two decimal places. It gives a false sense of precision of the measurements. I think one decimal place will suffice.

RESPONSE: These are excellent points. We have made the requested revisions to Table 1.

3. For the subsection on ‘Average head impact magnitudes by levels-of-contact and position group’ (p. 11), the word ‘average’ is ambiguous. I suggest making is clearer that all the values are medians.

RESPONSE: Thank you for identifying this ambiguous term, we have changed average to median in this section. 

4. For Figure 2, please make it clear that the plots show median number of impacts per player. Presumably the error bars indicate the IQR. In the footnotes, please ensure that all p-values greater than 0.001 are reported with and equal symbol (i.e. not inequality symbol).

RESPONSE: Thank you for noticing this. We have made the suggested revisions in the legend and fixed the equal symbols.

5. For Figure 3, please make it clear that the plots show median number of impacts per player. Presumably the error bars indicate the IQR.

RESPONSE: Thank you for noticing this. We have made the suggested revisions in the legend.

6. For Figure 3, please make it clear that the plots show median number PLA and PRA. Presumably the error bars indicate the IQR.

RESPONSE: Thank you for noticing this. We have made the suggested revisions in the legend.

Reviewer #3: 

Minor Issues:

1. We would recommend including the research question and hypotheses in the Abstract.

RESPONSE: In addition to the objective, we have revised the abstract to include the aspect of our research question aiming to determine which drills had the greatest head impact exposure, subsequently stating our primary hypothesis. Thank you for the suggestion. 

2. We would also recommend giving the dates the mentioned rule-changes and recommendations for reducing head trauma frequency in high-school / college football were established (levels-of-contact, tackle football, kickoffs rules) for a better overview on the subject and current stand (lines 56-65 and 74-79).

RESPONSE: Thank you for these suggestions. We have added years to two of the initiatives mentioned in the introduction. 

3. The authors give the number of previous concussions in Table 1 (line 193), however, they do not describe in the manuscript the relevance of their presence or absence; we would recommend to also meet this topic of risk potentiation in repetitive head impacts.

RESPONSE: Indeed, previous concussion history was not included in the analysis as a covariate, since our outcome measure is head impact exposure rather than neurologic outcome. Yet, this is an important point to discuss; therefore, we added a short paragraph elaborating the potential intersection between concussion and subconcussive head impacts right before the clinical implication paragraph in the discussion. Thank you for the suggestion.

4. The authors also give in Table 1 (line 193) in detail the race and ethnicity of the players, although 88% of the players are white and only 4% American Indian and 8% multiracial. Moreover, they do not explain in the manuscript potential interracial differences in concussion risk. We would recommend to either add this information or to shorten the table regarding this aspect.

RESPONSE: Excellent point. Reviewer #1 also pointed this out and we expanded the limitation section to acknowledge our lack of racial and ethnical diversity. We decided to keep this race/ethnicity in the table so that future research is encouraged to settle this unsolved question. 

Reviewer #4:

Material and Methods

line 123: this is a short season from mid-August 2019 to end of October 2019 for preseason training to the end of season?

RESPONSE: This is a great question. It was shorter season than expected because they lost out of the season earlier than expected on Friday, November 1st. Had they won a playoff game, the season could have lasted more weeks. There first varsity game of the season was on Friday, August 23rd.

Results

line 197: the results are clearly describing the level of contact and head impact exposure. However, what is missing, is how much time the athletes were performing activities of the different contact levels. e.g. if most time was spent doing live contact, it is no wonder that most head impact was generated during this typ of impact. it would be optimal to also include exposure time...

RESPONSE: Thanks for pointing this out, we agree. We were sure to include this at the end of our limitations section and made an addition of “time spent in each level of contact.” 

A future direction for this line of research is to examine subject specific head impact data that also incorporates time, and more importantly repetitions participated in. Part of the issue with team-wide contact guidelines/restrictions is that there is such large variance between players in both practice and game play participation.

For example, as you can imagine, in any given hour of certain drill one player may experience 50 repetitions of drills and associated head impacts, whereas another player may experience only 0 to 10 repetitions. This creates a significant flaw for team-wide restrictions that are based on contact time rather than repetitions.

Discussion

line 231: also dependent on exposure time, see comment above...

RESPONSE: Thank you again, this is an important limitation of the study. See comment above too. Time is an important variable to consider and we would like to see it added as part of future research designs exploring intensity levels. Despite this limitation, and our previous comment above, ultimately the outcome of cumulative head impact exposure per level-of-contact is still an important finding even without the amount of time involved. For example, policy makers and other stakeholders should still be interested in which levels have the most head impact exposure associated with them and could potentially utilize those levels to decrease exposure.

line 233: the skill group received much less cumulative impacts 130 vs over 200. Although no statistically significant finding, don't you think that receiving 30% less impacts does not count? please discuss! otherwise please add power analysis and sample size calculation etc... please see also line 254 where you say so, that skill positions have leas head impact

RESPONSE: Thank you for pointing this out. It is, indeed, a noticeable difference in head impact experiences between linemen/hybrid vs. skill. There are some heterogeneity in within-group head impact exposure where some skill players sustain many hits while several linemen experienced less hits, which points to the issue of lack of sample size. We elaborated this issue (clinical significance vs. statistical significance) in both the 1st and 2nd paragraphs of the discussion. This is a great point and we appreciate your suggestion. 

line 299ff: another limitation is that exposure time of the different levels of contact is not reported...

RESPONSE: We added this revision more clearly in two statements within the limitations section. Thanks for your emphasis of this point.

line 318: usually references are not cited in the conclusion, since the conclusions are your main findings and not the findings of other studies (since you published an original study)

RESPONSE: Thanks for identifying this irregularity. We thought about this and it is our hope that we could leave these references in the conclusions as a reminder to the audience that how our results may play a catalytic role to fill the current gap in our knowledge. We would really like to hammer home the point as part of our conclusion that greater dissection of football structure and guidelines may lead to measures that more effectively decrease subconcussive head impact exposure. To balance your perspective, the conclusion section within the abstract does not contain such referential statement. We hope that you will understand our rationale.

Tables

Table 1:please remove psychiatric conditions, since there are none. makes the table smaller and easier to read... also remove the lines "playing position, n (%)", since this information is redundant. the cumulative impact count is actually "average cumulative impact count"? also for the peak linear and rotational acceleration?

RESPONSE: We have made these revisions and significantly simplified Table 1.

Reviewer #5: For me as a neurologist it is very important that efforts are made to reduce the cumulative exposure of subconcussional cerebral microtrauma in adolescent high school football players. To verify the validity of the concept of different levels-of-contact which has been establishes by USA Football, the authors present an interesting study that compares the real exposure to mild cranial trauma at different levels-of-contact and different positions of football players. In this respect, the authors prove and validate the intuitive assumption that the different increasing levels-of-contact reflect a hierarchy of cumulative exposure to hits to the head. I have no objections to or suggestions for improvement of the manuscript.

RESPONSE: Thank you for reviewing our article, we appreciate your feedback and interest from neurologist standpoint. This is encouraging for us to hear.

---

## [Decision Letter · Decision Letter 1]

29 Jul 2020

PONE-D-20-13120R1

Subconcussive head impact exposure between drill intensities in U.S. high school football

PLOS ONE

Dear Dr. Kawata,

Thank you for submitting your manuscript to PLOS ONE. After careful consideration, we feel that it has merit but does not fully meet PLOS ONE’s publication criteria as it currently stands. Therefore, we invite you to submit a revised version of the manuscript that addresses the points raised during the review process.

ACADEMIC EDITOR:

- Please remove the references from the conclusion as this is uncommon. I also would advise to remove the lines 413 to 415 as it does not reflect the results of your study. Please also refer to the suggestions made in the review concerning the conclusion and adapt the abstract.

-I would ask you to add a short explanation for the physical denotations used (like "g" ..and krad/s2) so that interested readers (as non-concussion experts) get the channce to understand those values better. I think it is important to refer  to the "normal/usual values" (you can give examples) of forces applied to the head/brain in a normal envirment and in American Football. This might be helpful to sharpen the awareness of the medical staff. If you have any further questions in this topic you can contact me directly through the editorial manager to speed up your publication.

We look forward to receiving your revised manuscript.

Kind regards,

Markus Geßlein

Academic Editor

PLOS ONE

Reviewers' comments:

Reviewer's Responses to Questions

**Comments to the Author**

1. If the authors have adequately addressed your comments raised in a previous round of review and you feel that this manuscript is now acceptable for publication, you may indicate that here to bypass the “Comments to the Author” section, enter your conflict of interest statement in the “Confidential to Editor” section, and submit your "Accept" recommendation.

Reviewer #1: All comments have been addressed

Reviewer #2: All comments have been addressed

Reviewer #3: All comments have been addressed

Reviewer #4: All comments have been addressed

2. Is the manuscript technically sound, and do the data support the conclusions?

Reviewer #1: Yes

Reviewer #2: Yes

Reviewer #3: Yes

Reviewer #4: Yes

3. Has the statistical analysis been performed appropriately and rigorously? 

Reviewer #1: Yes

Reviewer #2: Yes

Reviewer #3: Yes

Reviewer #4: Yes

4. Have the authors made all data underlying the findings in their manuscript fully available?

Reviewer #1: Yes

Reviewer #2: Yes

Reviewer #3: Yes

Reviewer #4: Yes

5. Is the manuscript presented in an intelligible fashion and written in standard English?

Reviewer #1: Yes

Reviewer #2: Yes

Reviewer #3: Yes

Reviewer #4: Yes

6. Review Comments to the Author

Reviewer #1: The authors have done well to comply with the requested revisions. There are no further points of criticism.

Reviewer #2: Thank you for the opportunity to review the revised manuscript titled “Subconcussive head impact exposure between drill intensities in U.S. high school football”. The authors have satisfactorily addressed the reviewer comments. I only have one final suggestion.

P19L403-405: Please remove this sentence; it is not appropriate to include findings from previous studies in the conclusion section in this manner. I also recommend removing the words ‘Our data suggest that the…’ from the first sentence of the conclusion (P19L400), and replacing the words ‘Our data indicate the importance of considering…’ with ‘It is important to consider…’.

Reviewer #3: (No Response)

Reviewer #4: dear authors,

thanks you very much for the performed revision, all my questions have been fully answered, i have no further issues!

7. PLOS authors have the option to publish the peer review history of their article (what does this mean?). If published, this will include your full peer review and any attached files.

Reviewer #1: **Yes: **Juergen Taxis

Reviewer #2: No

Reviewer #3: No

Reviewer #4: No

---

## [Author Response · Author response to Decision Letter 1]

29 Jul 2020

Thank you for your thorough attention to important details. As pointed out, the conclusion section was revised to reflect reviewers’ phrasing as well as removal of references and its sentence all together, as suggested by the editor. Abstract’s conclusion section also reflects these changes.

Additionally, in the first paragraph of result section, we inserted some comparative information about g-force using an example of car crash, which is often used as a referential example in head trauma research. However, there are no good real-life examples about rotational acceleration (krad/s2), and this rotational acceleration is almost always correlated to g-force in concussion research (if player experience high, cumulative g-force, then their rotational acceleration (krad/s2) is high. Therefore, the car crash example for g-force should suffice to sharpen awareness of the medical personnel. 

Thank you.

---

## [Editor Report · Decision Letter 2]

4 Aug 2020

Subconcussive head impact exposure between drill intensities in U.S. high school football

PONE-D-20-13120R2

Dear Dr. Kawata,

We’re pleased to inform you that your manuscript has been judged scientifically suitable for publication and will be formally accepted for publication once it meets all outstanding technical requirements.

Kind regards,

Markus Geßlein

Academic Editor

PLOS ONE

Additional Editor Comments (optional):

Thank you very much for your thorough scientific work and your prompt cooperation.

Best regards

Reviewers' comments:

none

---

## [Editor Report · Acceptance letter]

6 Aug 2020

PONE-D-20-13120R2 

Subconcussive head impact exposure between drill intensities in U.S. high school football 

Dear Dr. Kawata:

I'm pleased to inform you that your manuscript has been deemed suitable for publication in PLOS ONE. Congratulations! Your manuscript is now with our production department. 

Kind regards, 

on behalf of

Dr. Markus Geßlein 

Academic Editor

PLOS ONE